# Preparation of Active Peptides from *Camellia vietnamensis* and Their Metabolic Effects in Alcohol-Induced Liver Injury Cells

**DOI:** 10.3390/molecules27061790

**Published:** 2022-03-09

**Authors:** Lu Feng, Jian Chen, Wuping Yan, Zhouchen Ye, Jing Yu, Guanglong Yao, Yougen Wu, Junfeng Zhang, Dongmei Yang

**Affiliations:** 1Key Laboratory of Food Nutrition and Functional Food of Hainan Province, Engineering Research Center of Utilization of Tropical Polysaccharide Resources, Ministry of Education, College of Food Science and Technology, Hainan University, Haikou 570228, China; scfenglu@163.com (L.F.); chenjian19850702@163.com (J.C.); 2College of Horticulture, Hainan University, Haikou 570228, China; ywp1734@163.com (W.Y.); 1196247235@163.com (Z.Y.); 992928@hainanu.edu.cn (J.Y.); zhangjunfengvip@163.com (J.Z.); ydm5711@126.com (D.Y.)

**Keywords:** *Camellia**vietnamensis* active peptide, purification, alcoholic liver injury, metabolomics

## Abstract

Liver damage seriously affects human health. Over 35% of cases of acute liver damage are caused by alcohol damage. Thus, finding drugs that can inhibit and effectively treat this disease is necessary. This article mainly focuses on the effect of the metabolome physical activity of active peptides in *Camellia vietnamensis* active peptide (CMAP) and improving liver protection. DEAE Sepharose FF ion-exchange column chromatography was used in separating and purifying crude peptides from *Camellia vietnamensis* Two components, A1 and A2, were obtained, and the most active A1 was selected. Sephadex G-100 gel column chromatography was used in A1 separation and purification. Three components, Al-1, Al-2, and Al-3, were obtained. Through antioxidant activity in vitro as an index of inspection, the relatively active component A1-2 was removed. Reverse-phase high-performance liquid chromatography showed that the purity of component A1-2 was 93.45%. The extracted CMAPs acted on alcoholic liver injury cells. Metabolomics studies revealed that the up-regulated metabolites were ribothymidine and xanthine; the down-regulated metabolites were hydroxyphenyllactic acid, creatinine, stearoylcarnitine, and inosine. This study provides an effective theoretical support for subsequent research.

## 1. Introduction

Camellia is the largest genus of the Theaceae. There are numerous species of Camellia, with an estimated 250 species all over the world. According to an analysis, it was reported that the protein content of the seed of Camellia is about 6–8%. The amino acid composition of *Camellia vietnamese* is about 17 kinds of essential amino acids representing the human body. Active peptides play a critical physiological effect on the human body. This paper aimed at excavating the effect of active peptides on alcoholic liver injury to uncover the relationship between the active peptides and severity of injury [1].

In the process of separating active peptides, multiple separation methods are usually combined. The cashew nut protein powder produced in Hainan was used as the raw material for preparing angiotensin-converting enzyme (ACE) inhibitory peptides through an enzymatic hydrolysis technology, and the structures of the separated main components were identified. The results showed that the optimal enzymatic hydrolysis conditions were 70 g/L substrate concentration and 1% enzyme. The M2 component with the highest ACE inhibitory peptide activity was identified at pH of 6.5, time of 7 h, and temperature of 55 °C. The amino acid sequence was Gly-Arg-Phe [2]. Extracted CMAPs were used in the cell experiments for analyzing the role of *Camellia vietnamensis* active peptides (CMAPs) in liver cells with alcoholic liver injury through metabolomics.

Metabolomics has been carried out in the diagnosis and prognosis of alcohol-induced liver disease (ALD) and research has been conducted on the pathogenesis of ALD and treatment effects [3]. The accumulation of free fatty acids in the liver may lead to pathological changes in ALD. Alcohol can significantly increase the active oxygen contents of L-02 cells and the expression levels of Nrf2 and p62 in the cells, which induce SOD and glutathione (GSH) activities in the cells. Identifying biomarkers is a key step in mass spectrometry-based metabolomics. Ferando et al. [4] used 2-DE and MALDI-TOF/TOF mass spectrometry techniques to identify differentially expressed proteins in the liver of rats fed with 5% (*v*/*v*) ethanol Lieber-DeCarli diet for 1 or 3 months. The results showed that the differential expression caused by ethanol is involved in multiple metabolic pathways, such as alcohol metabolism, lipid metabolism, and amino acid metabolism. Alcohol dehydrogenase (ADH) was down-regulated 1.6 times after 1 month, and aldehyde dehydrogenase (ALDH) was up-regulated 2.3 times after 3 months. By contrast, betaine-homocy steinem thytrasrse2 (BHMT2) was up-regulated 1.4 times at the same period, which is responsible for methionine metabolism, and peroxidase-1 expression was down-regulated 1.5 times at 3 months. Meanwhile, d-d-dopachromtautomerase was identified as a possible marker of ethanol-induced early steatohepatitis. Aroor et al. [5,6] reported that earbonicanyrse-I and guahionesm transferees muisoform levels in the liver of rats that had long-term or long-term binge drinking of alcohol were down-regulated. The prototype levels of protein disulfide isomerase associated protein 3 (PDIA3) in the liver of rats exposed to long-term alcohol consumption decreased significantly, whereas the level of acid PDIA3 increased. The level of glutamine synthetase (GS) in the liver of long-term drinking rats decreased. Obviously, GS can be used as a sensitive marker of peripheral liver injury. From the perspective of metabonomics, we further analyzed the metabolic pathways by which CMAPs inhibit liver injury.

## 2. Results

### 2.1. Separation by Ultrafiltration of Crude Camellia vietnamensis Active Peptides

As shown in Table 1, the molecular weight of the ultrafiltration membrane gradually decreased, the ACE inhibition rate of the samples gradually increased, and the IC_50_ value decreased. Some long peptides and impurities that had no ACE inhibitory activities were removed by ultrafiltration classification, and ACE inhibitory peptides were enriched. Korhonen et al. [7] used ultrafiltration to separate ACE inhibitor peptides in pepsin hydrolyzate from whey protein and found that a 1 kDa permeate had the smallest IC_50_ value. Enriched with liquid, the component with the highest ACE inhibition rate was the permeate with a molecular weight of less than 1 kDa, and the IC_50_ value of the lyophilized peptide powder was the smallest, indicating it had the highest activity, consistent with the results of the present study. Therefore, in this experiment, the ultrafiltration membrane was used in separating crude peptides from CMAP with a molecular weight of less than 1 kDa.

### 2.2. Ion Exchange Column DEAE-Sepharose F.F Separation and Purification of Crude Camellia vietnamensis Active Peptides

After the ionic concentration and the pH value of the starting buffer of the ionic column were determined through the test tube pre-experiment, crude *Camellia vietnamensis* meal peptide with a molecular mass of less than 1 kDa was obtained by membrane ultrafiltration and subjected to 0.25 and 0.35 mol/L NaCl. After elution with 10 mmol/L of Tris-HCl buffer (pH 8.0), two components, A1 and A2, were separated (Figure 1).

The IC_50_ values of the two components were measured. IC_50_-A1 = (0.145 mg/mL), IC_50_-A2 = (0.202 mg/mL). The reason that the IC_50_ value of the A1 component was lower than that of the A2 component may be that the A1 component contained more short-chain polypeptides and a greater proportion of free amino acids. Gel chromatography is usually used in separating and purifying ACE inhibitory peptides in *Theragra chalcogramma* Pallas; the IC_50_ reached 0.066 mg/mL and the final identification showed that the short-chain peptides were more abundant.

The A1 and A2 components were tested for their ability to remove DPPH, ·OH, and O_2_-· and total reduction ability, and the antioxidant effects of the components were evaluated. The measurement results are shown in Figure 2. Under the same concentration, although the antioxidant effects of the same sample in each test varied, the antioxidative effects of the two separated components were significantly different (*p* < 0.05), and the antioxidative effect of A1 was stronger. Concentration had a clear relationship, and a certain concentration dependence was observed. Through in vitro antioxidant activity screening, we found that the crude peptide A1 component of CMAP had strong activity. Therefore, a sufficient amount of A1 component was collected and freeze-dried for further separation and purification.

### 2.3. Separation and Purification of Camellia vietnamensis Active Peptides through G-100 Gel Chromatography

The CMAP peptides were separated using the DEAE-Sepharose FF ion-exchange column and dissolved at a concentration of 10 mg/mL. After desalting with a macroporous resin, 10 mL of each sample was loaded on a Sephadex G-100 column. Distilled water was used as the eluent, and the elution rate was 1.0 mL/min. After separation with a Sephadex G-100 gel, three different components were obtained, namely, A1-1, A1-2, and A1-3 (Figure 3).

By determining the components A1-1, A1-2, and A1-3 for the removal of DPPH capacity, OH removal capacity, O_2_ removal capacity, and total antioxidant capacity, the antioxidant effect of each component was assessed. The results of the measurement are shown in Figure 4. At a concentration of 10 mg/mL, the ability to remove DPPH and O_2_- and the total antioxidant capacity of component A1-2 were significantly higher than those of the other components (*p* < 0.05). This result showed that after a series of separation and purification, the main antioxidant components of CMAPs were concentrated in A1-2. Therefore, component A1-2 was selected for further separation studies.

### 2.4. Separation and Purification of Active Peptide A1-2 from Camellia vietnamensis Meal through RP-HPLC

RP-HPLC was used in separating and further purifying the active peptide A1-2 from *Camellia vietnamensis* meal, and its purity was determined. The result is shown in Figure 5. A single component peak was obtained, and the peaks of impurity were few. The other peaks of impurity were the peaks of solvents detected through the biuret method. The single component peak is the protein peak, and its peak area was 93.45%.

### 2.5. Discovery of Potential Biomarkers

As shown in Figure 6A, the gravel graph on the top left shows that the major components 1 (Dim-1: 41.5%) and 2 (Dim-2: 13.8%) can cumulatively explain nearly 55% of the variance. First and second principal components were sufficient to explain the trend in the data. Figure 6B shows that the indicators of high correlation between the main components 1 were L-octanoyl carnitine, oleoyl carnitine, and hexanoyl carnitine. Figure 6C shows that the distribution of the indices of the two main points was uniform, mainly L-phenylalanine, L-leucine, and nicotinamide. The effective ingredients were amino acids, and thus the active peptides in CMAP had some intervention effects on alcoholic liver damage.

The parameters of the model represent the interpretation rate of the model and the prediction rate of the model. Q^2^ and R^2^ are the quality evaluation indexes of the established PCA model, reflecting the difference of the prediction model. When Q^2^ > 0.5, it indicates that the prediction ability of the model is relatively strong. In this study, PCA was used to analyze the metabolic trajectories of five groups of samples and determine the classification and aggregation of the active peptides of camellia meal on alcohol-induced liver cell injury samples. 

From the PCA score chart (Figure 7), it can be seen that there is a significant separation between the ethanol group samples (C1, C2, and C3) and the control group samples (A1, A2, and A3) (Q^2^ = 0.856), indicating significant changes in metabolites. The alcohol-induced liver cell injury model was successfully demonstrated. The active peptide group samples (B1, B2, and B3) of the camellia meal and the control group samples (A1, A2, and A3) were also completely separated (Q^2^ = 0.986), indicating that, for the camellia meal, the addition of active peptides did not affect the metabolism of normal cells. For the samples of camellia meal (6 h) + ethanol group (D1, D2, and D3) and camellia meal (12 h) + ethanol group (E1, E2, and E3), the figure is clearly separated from the ethanol group (C1, C2, and C3) (Q^2^ = 0.574 and 0.607), indicating that the model has a good degree of fit and predictive ability.

### 2.6. Clustering Heat Map Analysis of Cell Metabolites

The heat map uses two colors to indicate the level of metabolites. Since the content of different metabolites varies greatly, all metabolites can first be standardized. The color bar on the right of the heat map indicates the value corresponding to the color, and the meaning of the value indicates the number of standard deviations from the average. The results of the cell metabolite clustering heat map analysis are shown in Figure 8 after the data were standardized by the unit variance method. This is an intuitive visualization method for analyzing the data distribution. It can be seen from Figure 7 that the content of metabolites in each group of cells is different. Acylcarnitines, amino acids, glycerophosphocholines, purines, and glycerophosphoethanolamines contain high contents of the five metabolomes. The cells in the control group and the ethanol group can be clustered separately, indicating certain differences in cell metabolites between the control group and the ethanol group. The camellia meal + ethanol group samples are clustered into the control group, indicating that the cells of the camellia meal group are given the camellia meal in advance under the protection of the active peptide. When ethanol is given to the injured samples, the metabolites are different from the ethanol group and tend to be similar to the control group.

After the data were screened for differences, the blank group was used as the control in Wayne analysis for the up-regulated and down-regulated metabolites of the other four groups of cell sample metabolites, as shown in Figure 9. Two differential metabolites were up-regulated, and four differential metabolites were down-regulated. The up-regulated metabolites were ribothymidine and xanthine. The down-regulated metabolites were hydroxyphenyllactic acid, creatinine, stearoylcarnitine, and inosine. Specific metabolites are available in Appendix A.

### 2.7. Analysis of Metabolic Pathways

The metabolic pathways of KEGG were analyzed, and MetaboAnalys was used. The diagrams of the metabolic pathways that can be affected by different metabolites between the CMAP + addition of ethanol and white groups (Figure 10) include: alanine metabolism; amino sugar metabolism; arachidonic acid metabolism; arginine and proline metabolism; tryptophan metabolism; aspartic acid metabolism; tricarboxylic acid cycle; glucose-alanine cycle; glycine and serine metabolism; purine metabolism; tyrosine acid metabolism; valine, leucine, and isoleucine degradation; and the urea cycle.

## 3. Discussion

In this experiment, the camellia lee protease hydrolyzate prepared by the best technology was ultrafiltrated using ultrafiltration membranes with different molecular weight sections. Three components were obtained, and the rate of ACE inhibition and IC_50_ value of each component were compared [6]. The results in Table 1 showed that the component with the highest ACE inhibition rate was the permeate with a molecular weight of less than 1 kDa and the lyophilized peptide powder had the lowest IC_50_ value but the highest activity. The results are similar to the findings of Korhonen and Pihlanto [7], who found that marine peptides with a relative molecular mass of less than 1 kDa have high antioxidative properties. Therefore, in this study, fractions with molecular weights of less than 1 kDa were collected and lyophilized for the subsequent separation step.

A DEAE Sepharose FF ion-exchange column chromatography system was used in separating and purifying crude peptides. Two components were obtained: A1 and A2 (Figure 1). Antioxidant activity was used as a screening guide, and the active component A1 (Figure 2) [1,8]. A Sephadex G-100 gel column chromatography system was used in separating and purifying A1. A1-1, A1-2, and A1-3 were obtained (Figure 3). Antioxidant activity in vitro was used as an indicator for inspection. The relatively strong component A1-2 was removed (Figure 4). Reverse-phase high-performance liquid chromatography results showed that the purity of component A1-2 was as high as 93.45% (Figure 5).

The mechanisms of protection of CMAPs on alcoholic liver damage cells were further studied [9,10,11]. UPLC-MS technology was used in examining the biomarkers and metabolic pathways of L-02 cell metabolism. The compositions of small molecular metabolites in cells were complex, and many differential metabolites were found. A total of 24 types of potential biomarkers related to metabolic pathways, including amino acids, metabolic derivatives of fatty acids, and carbohydrates, were detected (Figure 6). The most identified types were metabolites (38), and the liver is the only organ that can metabolize all amino acids; therefore, changes in the metabolism of amino acids are of great importance. The results of this study showed that the expression of the free aromatic amino acids (phenylalanine, tryptophan, and tyrosine) was up-regulated in the ethanol group compared with those in the white group, whereas the branched-chain amino acids (acid 4-guanidinobutyric and L-isoleucine acid) were down-regulated. This finding is consistent with the findings of previous studies [12,13]. Liver damage can lead to changes in amino acid metabolism and is often manifested as a decrease in the levels of free branched-chain amino acids and an increase in free aromatic amino acids. Branched-chain amino acids can inhibit the passage of aromatic amino acids through the blood–brain barrier, reduce ammonia content in the blood, and maintain nitrogen balance in the body. They can exert protective effects against liver damage [14,15]. Therefore, the protective effect of CMAPs on liver damage may be related to the up-regulation of free aromatic amino acids (phenylalanine, tryptophan, and tyrosine) and agmatine metabolism. CMAPs alter the contents of cellular metabolites in acute alcoholic liver injury mainly through fatty acid metabolism (increased level of buxin); amino acid metabolism by increasing free aromatic amino acids (phenylalanine, tryptophan, and tyrosine acid) and lowering the level of branched-chain amino acids (4-guanidinobutyric acid, L-isoleucine); carbohydrate metabolism (reduction in gluconic acid levels); ketoacid metabolism (increased levels of oxoglutarate); the metabolic pathway of purine (increased level of pseudouracil); and the metabolic pathway of pyrimidine (increased level of dihydrouracil). CMAPs can drastically down-regulate the levels of 4-guanidinobutanoic acid, L-isoleucine, and gluconic acid in the ethanol group and restore them to normal levels.

## 4. Materials and Methods

### 4.1. Chemicals and Reagents

*Camellia vietnamensis* T.C. Huang ex Hu was produced in Hainan. The following solutions were used: Tris-HCl (pH 8.0) buffer (Novozymes, Bagoswald, Denmark), NaCl (AR) (Novozymes, Bagoswald, Denmark), phosphate buffer (pH 7.4) (Novozymes, Bagoswald, Denmark), ferrous sulfate (AR) (Novozymes Bagoswald, Denmark), H_2_O_2_ (AR) (Novozymes, Bagoswald, Denmark), pyrogallol (AR) (Macklin, Shanghai, China), phenanthroline (AR) (Macklin, Shanghai, China), trifluoroacetic acid (AR), trypsin (250 U/mg) (Macklin, Shanghai, China), Sephadex G-100, acetonitrile (HPLC grade), methanol (HPLC grade) (Servicebio, Haikou, China), phosphate-buffered saline (PBS) solution (Servicebio, WuHan, China), transfer buffer (Servicebio, WuHan, China), absolute ethanol (AR) (Servicebio), and DMEM culture medium (Servicebio, WuHan, China).

### 4.2. Separation of Crude Active Peptides from Camellia vietnamensis by Ultrafiltration and Detection of Its IC_50_ Inhibition Rate

After the crude peptides of CMAP were dissolved in ultrapure water, 10, 5, and 1 kDa polyethersulfone ultrafiltration membranes were used for ultrafiltration. Each component was collected and lyophilized, and the IC_50_ value was determined. The highest ACE inhibition rate was selected. The component with the smallest IC_50_ value was purified [16,17].

HHL solution (100 μL) with a concentration of 5 mmol/L (dissolved in boric acid buffer containing 0.6 mol/L NaCl, pH 8.3) was added to 50 μL of *Camellia vietnamensis* enzymatic hydrolysis solution, and then incubated in a constant temperature water bath at 37 °C for 5 min. Next, 250 μL of 1 mol/L HCl was added immediately after 50 μL of ACE was added to it for 30 min to stop the reaction; Then, 1.5 mL of cold ethyl acetate was added to it, vortexed to mix evenly, centrifuged at 4000 r/min for 10 min, and 1.0 mL of the ester layer was transferred to another test tube. After fully drying, it was redissolved in 3.0 mL of deionized water, and the absorbance A was measured at 228 nm. At the same time, 50 μL of buffer was used instead of buffalo milk enzymatic hydrolysate as a blank control group; 50 μL of sodium borate buffer was used as a blank group of samples instead of ACE solution. $\;\mathrm{ACE}\;\mathrm{inhibition}\;\mathrm{rate}\left(\%\right)=\frac{A_{1}-A_{2}}{A_{1}-A_{3}}\times 100\%$. In the formula, A1 is the blank control group, A2 is the sample group, and A3 is the blank group of the sample.

### 4.3. DEAE Sepharose F.F. Ion Exchange Column Separation and Purification of Crude Camellia vietnamensis Active Peptide

A 1.6 m × 30 cm glass chromatography column was attached to an iron frame and the lower water outlet pipe was tightened. Approximately 5 mL of Tris-HCl buffer (pH 8.0) was injected at 10 mmol/L. The gel was added to the chromatography column slowly, continuously, and evenly, and the water outlet was opened. After the gel surface was 4 cm from the top of the chromatography column, the constant flow pump was turned on, and the column was equilibrated two times with a column volume buffer for the stabilization of the column bed. Approximately 10 mL of the dried sample with a concentration of 5 mg/mL was collected after ultrafiltration, dissolved in 10 mmol/L of Tris-HCl buffer (pH 8.0), and filtered with a 0.45 μm membrane filter. A preparative chromatography column was then prepared. The liquid phase underwent linear gradient elution. According to the results of the linear elution, the NaCl concentrations of the gradient eluate were 0.25 and 0.35 mol/L. The components A1 and A2 were collected according to the chromatographic peaks, and salt was removed through dialysis and lyophilization. The antioxidant effect of each component was determined. The antioxidative effect was evaluated through the following methods: Determination of the DPPH ability of camellia meal polypeptide samples. Referring to the method of Dawidowicz et al. [18], 0.5 mL of 1 mg/mL sample solution was added to 2.5 mL of 0.1 mmol/L DPPH·absolute ethanol solution, and the absorbance was measured at 517 nm after being placed in the dark for 30 min, denoted as B_1_; blank. In the experimental group, 0.5 mL of anhydrous ethanol was used to replace the sample solution. The other operations remained unchanged, and the absorbance value was recorded as B_0_. The formula is as follows: $\;\mathrm{clearance}\;\mathrm{rate}=\frac{B_{0}-B_{1}}{B_{0}}\times 100\%$.

Determination of scavenging OH ability of camellia meal polypeptide samples. Slightly modified from the assay used by Olaniyi et al. [19]. After 2 mL of 0.15 mol/L phosphate buffer (pH 7.4) and 1 mL of 1 mg/mL sample solution were sequentially added to 1 mL of 0.75 mmol/L phenanthroline solution prepared in absolute ethanol, the solution was immediately vortexed to mix uniformly. Next, 1 mL of 0.75 mmol/L ferrous sulfate solution was added. Finally, 1 mL of 0.1 mL/L H_2_O_2_ solution was added. After a water bath at 37 °C for 60 min, the absorbance value Ds at a wavelength of 536 nm was measured. The absorbance value Dc of ultrapure water was measured instead of the sample and H_2_O_2_ solution. The absorbance value Db of ultrapure water and H_2_O_2_ solution was measured instead of the sample and H_2_O_2_ solution. The formula is as follows: $\mathrm{clearance}\;\mathrm{rate}=\frac{\mathrm{Ds}-\mathrm{Db}}{\mathrm{Dc}-\mathrm{Db}}\times 100\%$.

Determination of the ability of camellia meal polypeptide samples to scavenge O_2_-. Based on a slightly modified method by Petropoulos et al. [11], 100 μL of 1 mg/mL sample solution was taken by pipette and added to 2.8 mL of 0.1 mol/L Tris-HCl (pH 8.2) buffer and vortexed. Then, the sample was homogenized and heated in a water bath at 25 °C for 10 min. Following this, 0.1 mL of 3 mmol/L of pyrogallol solution was added, immediately mixed thoroughly, and the absorbance (A) at 325 nm was measured. The data were recorded every 30 s for 5 min and ended at 5 min. As a linear regression equation of the change of absorbance A with time, the oxidation rate of pyrogallol is the linear slope Vs, and the oxidation rate of the blank control group with ultrapure water instead of the sample is the linear slope Vc, and the calculation formula is as follows: $\;\mathrm{clearance}\;\mathrm{rate}=\frac{\mathrm{Vc}-\mathrm{Vs}}{\mathrm{Vc}}\times 100\%$.

Determination of total antioxidant capacity of camellia meal polypeptide samples. According to the reagent instructions of the total oxidative capacity detection kit, the principle is to use substances with an antioxidant effect to reduce Fe^3+^ to Fe^2+^. The phenanthroline substances and Fe^2+^ form a stable compound together, and the absorbance value A is determined by colorimetry. This reflects the strength of the antioxidant capacity.

### 4.4. Separation by Chromatography on a Sephadex G-100 Column and Purification of Crude Camellia vietnamensis Active Peptides

Sephadex G-100 gel was used for separation. After gel pretreatment, it was added to a 2 cm × 100 cm chromatography column and equilibrated for 3–14 h. The flow rate was reduced to 1 mL/min before the sample was loaded (component A1 was dissolved in distilled water and had a solubility of 10 mg/mL; the loading volume was 10 mL) and eluted with distilled water. The detection wavelength was 280 nm, and the absorption peaks obtained were collected. The collected solutions were concentrated and lyophilized, stored under appropriate conditions, and dialyzed for salt removal. The antioxidative effects of the three obtained components were analyzed [20,21,22].

### 4.5. Preparation of RP-HPLC Chromatographic Separation and Purification of Camellia vietnamensis Active Peptides

The lyophilized powder, which was the component A1-2 with the highest antioxidant activity and separated by Sephadex G-100 gel chromatography, was dissolved in ultrapure water, filtered through a 0.22 mM membrane filter, and further separated and purified through preparative RP-HPLC chromatography. The following conditions were used: column, C18 column (φ4.6 × 250 mm, 5 mM); detection wavelength, 215 nm; mobile phase A, acetonitrile; and mobile phase B, an ultrapure aqueous solution containing 0.10% trifluoroacetic acid. The elution conditions were as follows: 0–5 min, 0% A; and 5–30 min, 0–40% A. The loading volume was 10 μL, and the column temperature was 35 °C. The purity of the synthetic peptides was analyzed using a Shimadzu Inertsil ODS-SP C18 column (4.5 mm × 250 mm, 5 mM), and the final purity was greater than 98%.

### 4.6. Cell Grouping and Processing

Normal human L-02 hepatocytes in the logarithmic growth phase were used. After trypsin digestion, a single cell suspension with DMEM medium containing 10% serum was inoculated into a 96-well cell culture plate at 37 °C and grown for 24 h in 5% CO_2_ environment. The 200 μg/mL active peptides from CMAPs and 200 mM ethanol were selected for metabolomics analysis, and the following groups were established: (A) white group, normal culture for 30 h; (B) CMAP group: containing 200 μg/mL of CMAP, the active peptides were cultured for 30 h; (C) ethanol group, 200 mM ethanol for 30 h; (D) CMAP + ethanol group, first cultivated with 200 µg/mL active peptide for 6 h, then 200 mM ethanol for 24 h; (E) CMAP + ethanol (WB) group: 200 μg/mL active peptide + ethanol group (WB); first incubated at 200 μg/mL active peptide for 12 h, then with 200 mM ethanol for 24 h. After culturing, cell metabolites were collected for testing.

### 4.7. Sample Collection and Preparation

The medium was quickly discarded and washed twice with PBS precooled to 4 °C. The remaining PBS was aspirated, then 500 μL of 80% aqueous methanol solution frozen at −80 °C was added for 4–6 h to cover the entire cell surface. The cells were incubated at 80 °C for 20 min and subsequently scraped with a cell scraper and collected and transferred to 1.5 mL centrifuge tubes (Eppendorf tubes). Approximately 500 μL of an aqueous solution of methanol was added before the cells were collected again. The samples were vortexed for 30 s and centrifuged (4 °C × 12 kr/min × 10 min). The supernatant was filtered through a 0.22 mM membrane, transferred to another 1.5 mL tube, and evaporated to dryness under a stream of nitrogen. The samples were reconstituted in the initial mobile phase and detected by LC-MS [23,24,25].

### 4.8. Preparation of Quality Control Samples

Approximately 10 μL of each sample was mixed with the quality control samples. Before testing sample analysis, 10 quality control samples were tested continuously, and one quality control sample from every four samples was tested at random.

### 4.9. Test Conditions

ACQUITY UPLC HSS T3 1.8 mM, 2.1 mm × 100 mm columns (Waters, Dublin, Ireland) were used, and the Agilent 1290 II UPLC- instrument Sciex QTOF 5600 PLUS LC/MS was used in instrument and electrospray ionization (ESI) mode. The analysis conditions were as follows: ion spray voltage in positive ion mode, 5500 V; ion spray voltage in negative ion mode, −4500 V; temperature, 450 °C; atomization gas, 50 psi; and auxiliary gas, 50 psi. The internal standards were phenylalanine-D8, tryptophan-D8, isoleucine-D10, asparagine-13C4, methionine-D3, valine-D8, proline-D7, alanine-D4, glycine-D2, serine-D3, glutamate-D5, aspartate-D3, arginine-D7, glutamine-D5, lysine-D9, histidine-D5, and taurine-D2.

Raw MS data were inputted to Markerview (version 1.3, AB Sciex) for peak identification and alignment. On the basis of secondary mass spectrum data, standard mass spectrum data, Metablites (AB SCIEX), METLIN (https://metlin.scripps.edu/landing_page.php?pgcontent=mainPage, accessed on 6 December 2019), and HMDB (http: //www.hmdb.ca/, accessed on 6 December 2019) database to identify metabolites [26,27].

### 4.10. Data Processing and Statistical Analysis

Origin Pro 9.1, GraphPad Prism 5, SIMCA-P 13.0, MetaboAnalyst 3.0, and Cytoscape 3.7.0 were used for mapping. All experiments were performed three times and the data were expressed as the mean ± standard deviation ($\bar{x}=\pm s$). SPSS Statistics 19.0 Turkey–Kramer was used in analyzing significant differences among the data.

## 5. Conclusions

A DEAE Sepharose F.F ion-exchange column chromatography system was used in separating and purifying crude peptides from *Camellia vietnamensis*. Two components, A1 and A2, were obtained. Antioxidative activity was used as the screening guide, and the active component A1 was selected. A Sephadex G-100 gel column chromatography system was used in separating and purifying A1. Three components, A1-1, A1-2, and A1-3, were obtained. In vitro antioxidative activity as an indicator for inspection, and components A1-2, which were active, were screened out. Reversed-phase high-performance liquid chromatography and SDS-PAGE electrophoresis and detection showed that A1-2 had a single peak and single band and had a molecular mass of approximately 13 kDa and purity of 93.45%. The addition of CMAPs remarkably improved the metabolism of cells damaged by ethanol by considerably down-regulating the elevated levels of 4-guanidinobutanoic acid, L-isoleucine, and gluconic acid to normal levels. Metabolomics studies revealed that the up-regulated metabolites were ribothymidine and xanthine, and the down-regulated metabolites were hydroxyphenyllactic acid, creatinine, stearoylcarnitine, and inosine. Metabonomics studies revealed that CMAPs mainly regulate changes in small molecular metabolites in acute alcoholic liver injury cells via the amino acid, fatty acid, and carbohydrate metabolism pathways.

## Figures and Tables

**Figure 1 molecules-27-01790-f001:**
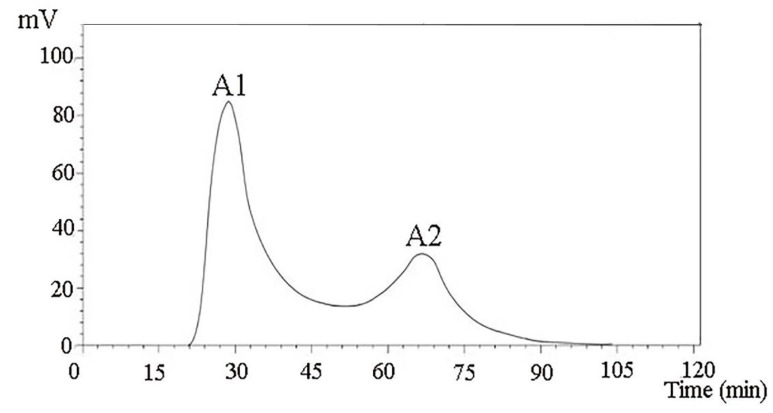
Purification spectra of peptides from *Camella vietnamensis* meal by DEAE Sepharose F.F.

**Figure 2 molecules-27-01790-f002:**
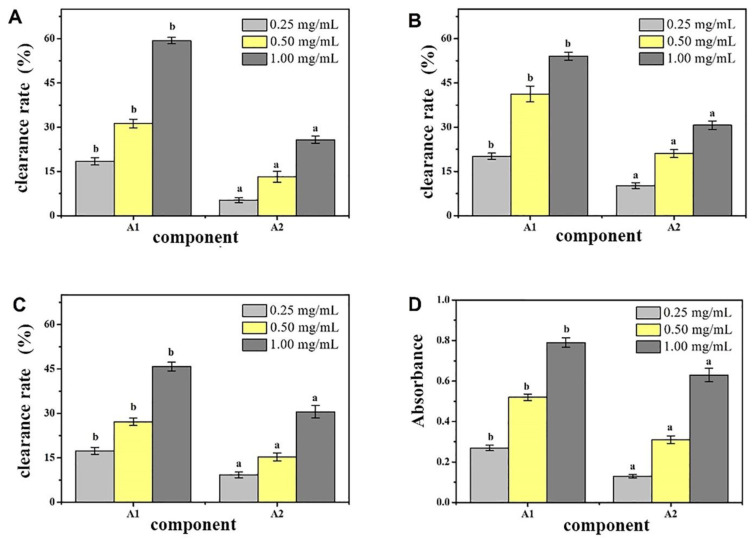
Antioxidant activity of different peptides fractions A1 and A2 from *Camella vietnamensis* in vitro. (Note: (**A**) was the scavenging effect on DPPH, (**B**) was the scavenging effect on ·OH, (**C**) was the total reducing power, and (**D**) was the scavenging effect on superoxide anion.)

**Figure 3 molecules-27-01790-f003:**
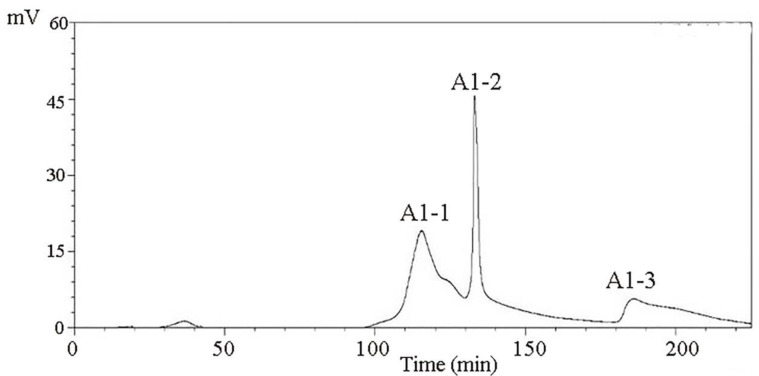
Purification spectra of peptides from *Camella vietnamensis* meal by Sephadex G-100.

**Figure 4 molecules-27-01790-f004:**
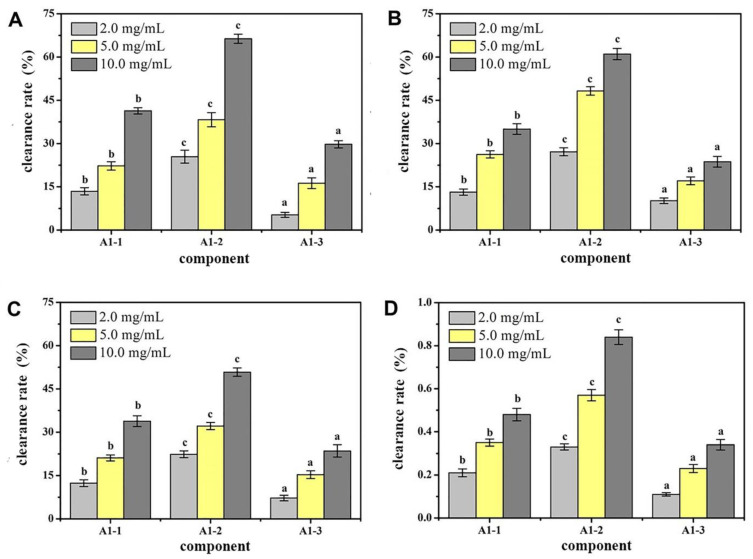
Antioxidant activity of different peptides fractions A1-1, A1-2 and A1-3 from *Camella vietnamensis* in vitro. (Note: (**A**) was the scavenging effect on DPPH, (**B**) was the scavenging effect on ·OH, (**C**) was the total reducing power, and (**D**) was the scavenging effect on superoxide anion.)

**Figure 5 molecules-27-01790-f005:**
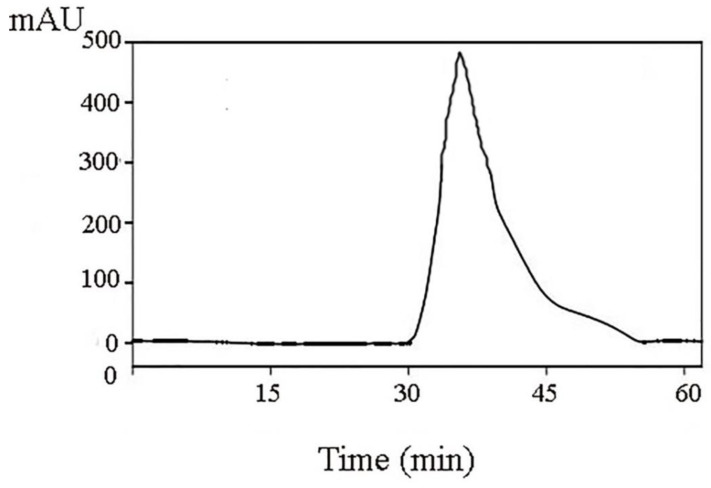
Purification of A1-2 by RP-HPLC.

**Figure 6 molecules-27-01790-f006:**
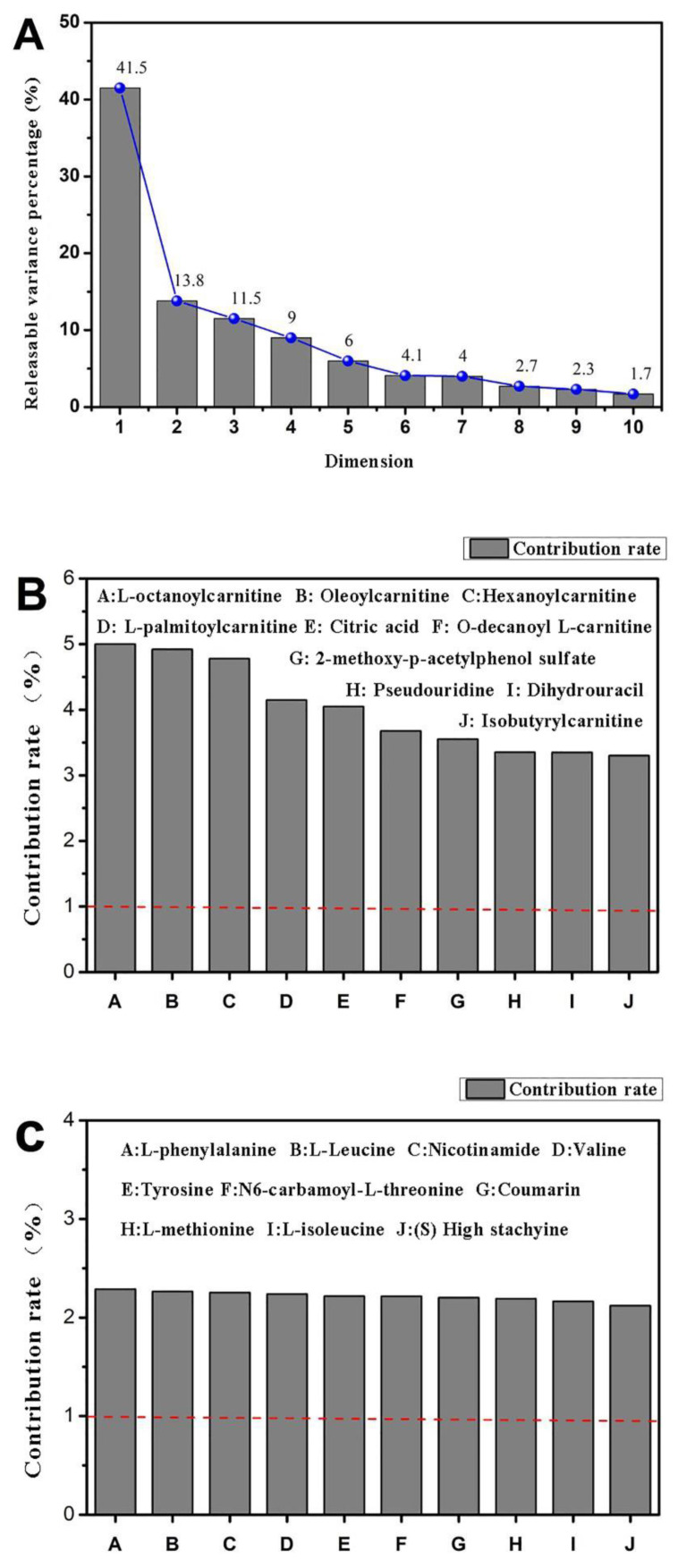
Scree plot of L-02 cell metabolites ((**A**): Interpretable variance for each principal component; (**B**): Metabolites in Principal Component 1; (**C**): Metabolites in Principal Component 2).

**Figure 7 molecules-27-01790-f007:**
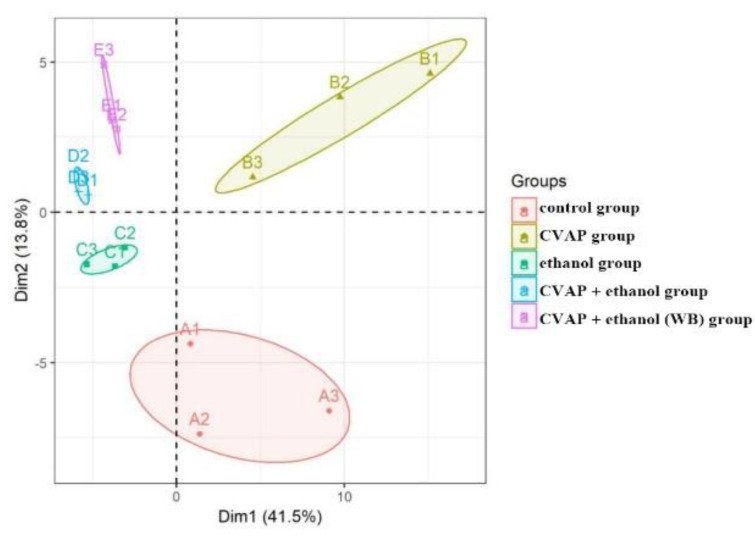
PCA score scatter plot of L-02 cell metabolites.

**Figure 8 molecules-27-01790-f008:**
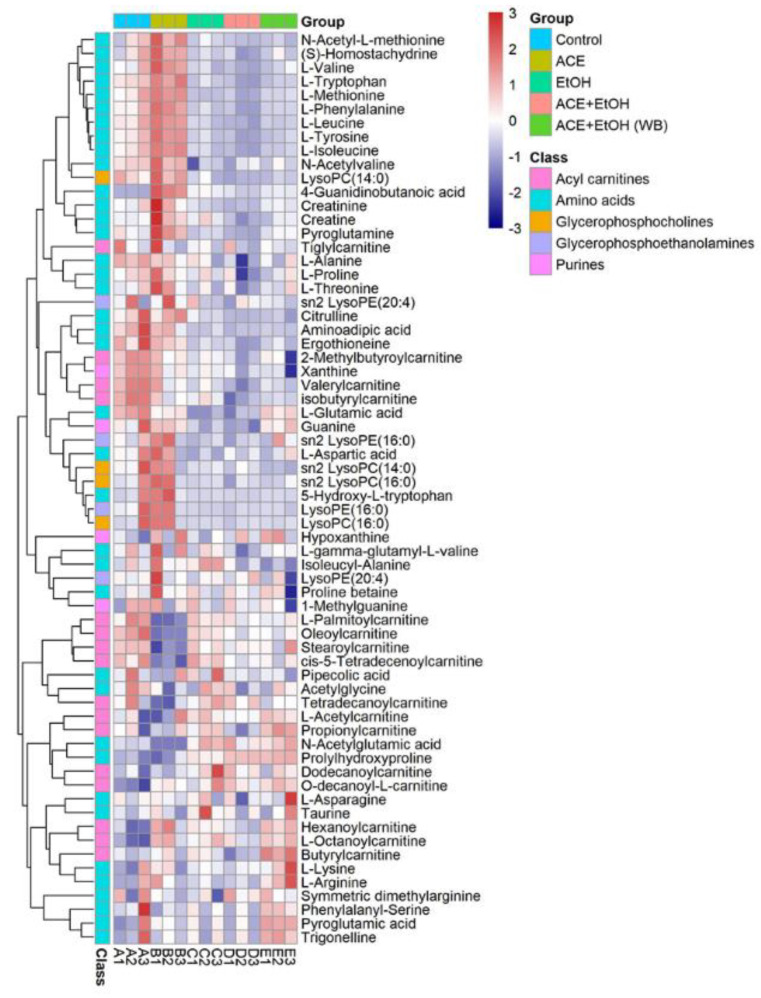
Clustered heatmap of L-02 metabolites.

**Figure 9 molecules-27-01790-f009:**
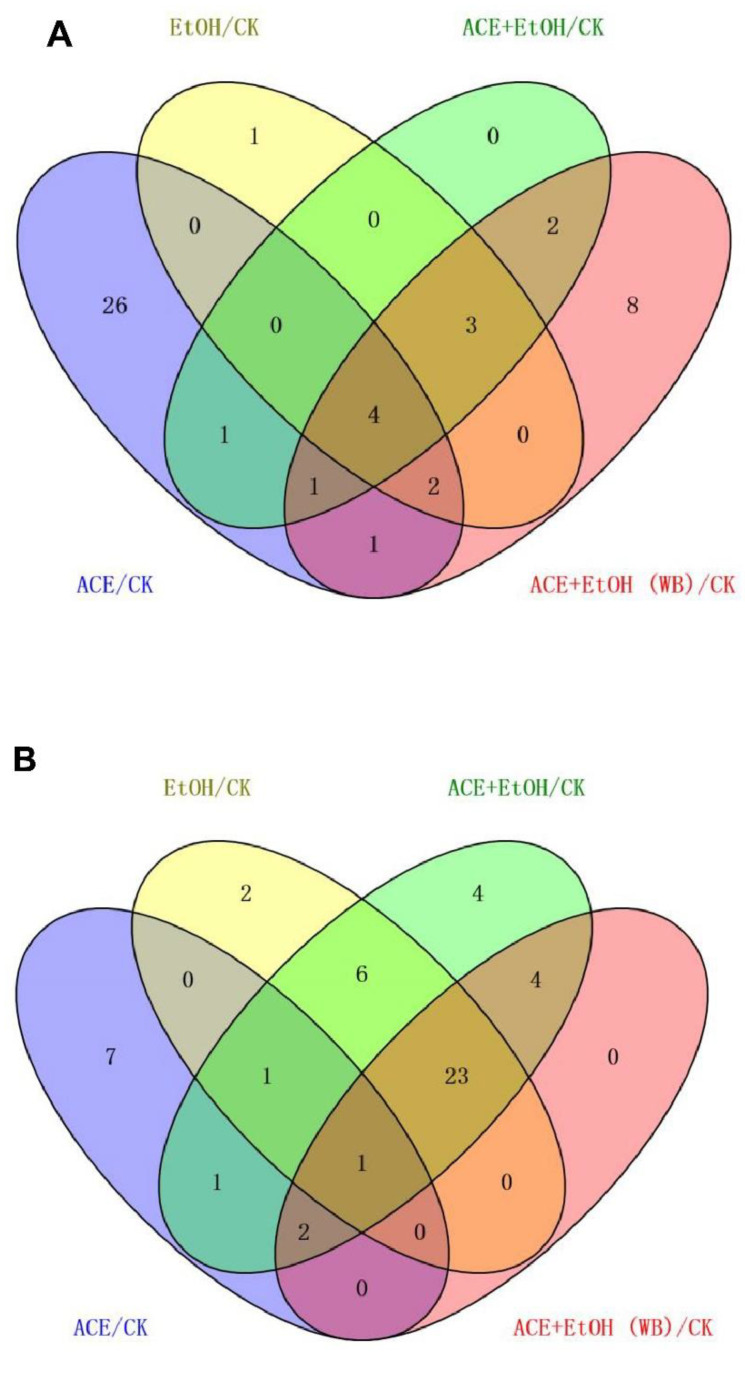
Venn diagram of differences among the groups ((**A**): up-regulated metabolites; (**B**): down-regulated metabolites).

**Figure 10 molecules-27-01790-f010:**
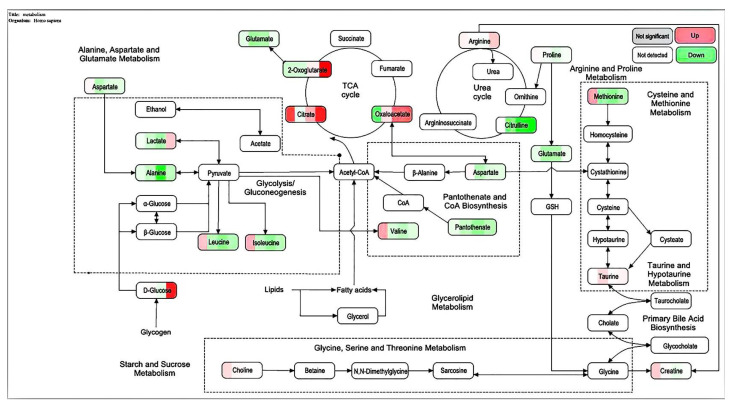
Bubble diagram of the importance of the metabolic pathway of L-02 cells and the roles of active peptides from *Camellia vietnamensis* meal and ethanol.

**Table 1 molecules-27-01790-t001:** ACE inhibition rate and IC_50_ of ultrafiltration fractions in *Camella vietnamensis* meal protease solution.

Molecular Mass (kDa)	ACE Inhibition Rate %	IC_50_/mg·mL^−1^
<10 kDa	73.78 ± 1.14	0.874
<5 kDa	77.27 ± 0.89	0.728
<1 kDa	79.24 ± 1.14	0.678

## Data Availability

Not applicable.

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
