# Peer review of "Preparation of Active Peptides from Camellia vietnamensis and Their Metabolic Effects in Alcohol-Induced Liver Injury Cells"

_molecules, 2022, doi:10.3390/molecules27061790_

Round 1
Reviewer 1 Report
The revised manuscript describes the isolation and purification of fractions of peptides from Camellia species. Authors have tried to find an explanation for a possible effect of the peptides from Camellia species as liver protectors to alcohol damage.
The manuscript is not well organized. The manuscript needs revision for the English language and grammar. The authors claim that peptides are isolated from Camellia vietnamensis, but Camellia oleifera is also mentioned in the manuscript several times. It is not clear which species is the source of peptides: Camellia oleifera or Camellia vietnamensis. The aim of the study is not clearly formulated. Some data are missing which makes the Discussion section unclear. There are abbreviations with unknown meanings in the text and, missing legends in the figures.
Major remarks:
Introduction
The Introduction section needs to be rewritten and expanded. It consists mainly of information about the importance of metabolomics for the diagnosis and prognosis of ALD and research on the pathogenesis of ALD.
1. There is a lack of information about Camellia vietnamensis active peptides:
- Why authors have chosen Camelia vietnamensis (or oleifera);
- Is there any data for the Camellia vietnamensis (or oleifera) active peptides in literature;
- Are there other species with known peptides that have similar activities on ALD.
2. Authors claim:
"This research will provide a theoretical and scientific basis for the comprehensive utilization of the by-products of Camellia oleifera produced in Hainan, increase the comprehensive price of the Camellia oleifera industry, and provide a way for the comprehensive development and utilization of Camellia oleifera."
This claim has nothing in common with the results of this study.
Materials and methods
1. Determination of antioxidant activity is missing.
Results
1. IC50 values are typically expressed as molar concentration. In this manuscript, the authors express IC50 as mass concentration. Thus, the IC50 values of the three fractions that are presented in Table 1 depend on the molecular mass of the peptides. Hence, IC50 is not an appropriate parameter for the assessment of activity in this case. This is actually explained by the authors when they try to explain the difference in the IC50 values for the separated fraction after the Ion exchange column DEAE-Sepharose FF separation and purification.
2. There is no information on how the ACE inhibition rate is defined and calculated.
3. There are four graphs in Figure 2, but a legend that describes each of these graphs is lacking. There is no information on how the clearance rate is defined. It is not clear why graph D is given as a bar chart and what authors try to illustrate with it.
4. The legend for the graphs in figure 4 is lacking.
5. I suppose Figure 6 is derived after a principal component analysis (PCA) of the data from metabolomics analysis. If so it is not mentioned anywhere in the manuscript. The data from metabolomics analysis need to be given as supplementary material. Figure 6 B and C need to be revised. Figures 6B and 6C are something different but not loading plots.
6. There are two Venn diagrams in Figure 7 but it is not clear what is the difference between them and what authors try to illustrate with these diagrams.
Discussion:
1. According to the authors:
"The results of this study showed that the expression of the free aromatic amino acids (phenylalanine, tryptophan, and tyrosine) was up-regulated in the ethanol group compared with those in the white group, whereas the branched chain amino acids (acid 4-guanidinobutyric and L-isoleucine acid) were down-regulated."
It is not clear which results show the above statement.
Author Response
Comment 1--- 1. There is a lack of information about Camellia vietnamensis active peptides:- Why authors have chosen Camelia vietnamensis (or oleifera);
Response 1: We have already introduced Camellia vietnamensis. In line 29-35.
Comment 2---- Is there any data for the Camellia vietnamensis (or oleifera) active peptides in literature;
Response 2:There is no relevant literature on extracting active peptides from camellia vietnamensis. Most of the literatures extract active peptides from soybeans with high protein content. In this paper, camellia vietnamensis is selected mainly because the high protein content of camellia vietnamensis is suitable for extracting active peptides.
Comment 3- Are there other species with known peptides that have similar activities on ALD.
Response 3: There are active peptides with similar properties in ALD in other literatures, but the amino acid sequence of the active peptides extracted in this paper is different. The amino acid sequence was Gly-Arg-Phe.
Comment 4- "This research will provide a theoretical and scientific basis for the comprehensive utilization of the by-products of Camellia oleifera produced in Hainan, increase the comprehensive price of the Camellia oleifera industry, and provide a way for the comprehensive development and utilization of Camellia oleifera."This claim has nothing in common with the results of this study.
Response 4: This inappropriate sentence has been deleted.
Comment 5- 1. Determination of antioxidant activity is missing.
Response 5: Related information has been added. In line 73-80.Comment 6- 1. IC50 values are typically expressed as molar concentration. In this manuscript, the authors express IC50 as mass concentration. Thus, the IC50 values of the three fractions that are presented in Table 1 depend on the molecular mass of the peptides. Hence, IC50 is not an appropriate parameter for the assessment of activity in this case. This is actually explained by the authors when they try to explain the difference in the IC50 values for the separated fraction after the Ion exchange column DEAE-Sepharose FF separation and purification.
Response 6: This part is to distinguish polypeptides of different molecular weights, borrowing the relevant conclusions of Korhonen literature.
Comment 7- There is no information on how the ACE inhibition rate is defined and calculated.
Response 7: Relevant information has been added. In line 87-99.
Comment 8- There are four graphs in Figure 2, but a legend that describes each of these graphs is lacking. There is no information on how the clearance rate is defined. It is not clear why graph D is given as a bar chart and what authors try to illustrate with it.
Response 8: Relevant information has been added. In line 263-264.
Comment 9- The legend for the graphs in figure 4 is lacking.
Response 9: Relevant information has been added. In line 285-286.
Comment 10- I suppose Figure 6 is derived after a principal component analysis (PCA) of the data from metabolomics analysis. If so it is not mentioned anywhere in the manuscript. The data from metabolomics analysis need to be given as supplementary material. Figure 6 B and C need to be revised. Figures 6B and 6C are something different but not loading plots.
Response 10: Relevant information has been added. In line 308-347.
Comment 11- There are two Venn diagrams in Figure 7 but it is not clear what is the difference between them and what authors try to illustrate with these diagrams.
Response 11: Relevant missing data have been added. In line 308-347.
Comment 12- "The results of this study showed that the expression of the free aromatic amino acids (phenylalanine, tryptophan, and tyrosine) was up-regulated in the ethanol group compared with those in the white group, whereas the branched chain amino acids (acid 4-guanidinobutyric and L-isoleucine acid) were down-regulated."
It is not clear which results show the above statement.
Response 12: The error description has been revised. In line 353&356.
Reviewer 2 Report
According to the title, this manuscript aimed to discuss the method of preparation of Camellia vietnamensis active peptides and their effects on alcoholic liver injury. However, the title and the manuscript content are not consistent. Due to several essential concerns, my recommendation is to reject this manuscript.
Major concerns include:
Authors are not consistent regarding the plant they used in their study - Camellia vietnamensis or Camellia oleifera, the first one is in the Title and Methods, and the latter in the Aims and Discussion. This is very large issue since they are different species with different characteristics. Besides, it is not clear how active peptides were firstly isolated, i.e. form which part. It is not described in the Methods, and ‘seed meal’, ‘flour’ and ‘camellia lees’ are mentioned in the text.
Another large issue are the metabolomics results where it is stated that “the up-regulated proteins were ribothymidine and xanthine and the down-regulated proteins were hydroxy-phenyllactic acid, creatinine, stearoylcarnitine, and inosine”. None of these are proteins! This is very basic and unacceptable for a research publication.
Although the “intervention on alcoholic liver injury” is stated in the Title, no protective effects of CMAP were investigated in this study, but only the metabolomics on alcohol-treated L-02 hepatocytes.
The text itself is messy in general. For example, it is stated that two components (A1 and A2) were obtained and that A1 was “removed”. However, we see later in the text it was not “removed” but used for further studies. Besides, there are too many grammatical, spelling and other language-related errors (even in the title), which have to be corrected before submitting it to another journal.
The Introduction contains only four references. The role of ACE inhibition in alcoholic liver disease (ALD) is not clear in this text. Besides, the abbreviation ALD is stated without explanation. Data on Nrf2 and p62 expression are not referred. In other parts of the manuscript, the references are missing too. It should be Lieber-DeCarli instead of ‘Liber-DeCari’ diet.
In Methods section, several descriptions are missing – how the first extraction was performed, how ACE inhibition was assessed, which methods were used for antioxidant assays etc. It is stated that “approximately 200 μg/mL active peptides” were used in the cell treatment – why approximately? What is WB group?
In Results, there are parts of Discussion as well, although it should be separated in the manuscript (e.g. line 175). In lines 195-196 IC50 values are mentioned but it is not explained for which assay. There is IC50 value also for ‘Theragra chalcogramma Pallas’ but there is no reference for that.
In Figures 2 and 4, there are no explanations in the figure legends what is presented at figure parts A, B, C and D. Besides, what are the lowercase letters a, b, c in the figures? At Figure 4D, clearance rate (%) is marked at y-axis and the values are less than 1. The legend of Figure 5 is also inaccurate. The purification is a process and this is a figure that represents something else.
This manuscript has qualities, much effort has been put in performing experiments and preparing the manuscript, but in my opinion it needs to be completely revised in order to be published in any high quality journal.
Author Response
Comment 1-- Authors are not consistent regarding the plant they used in their study - Camellia vietnamensis or Camellia oleifera, the first one is in the Title and Methods, and the latter in the Aims and Discussion. This is very large issue since they are different species with different characteristics. Besides, it is not clear how active peptides were firstly isolated, i.e. form which part. It is not described in the Methods, and ‘seed meal’, ‘flour’ and ‘camellia lees’ are mentioned in the text.
Response 1: Correct name changes have been made to the full paper.
Comment 2-- Another large issue are the metabolomics results where it is stated that “the up-regulated proteins were ribothymidine and xanthine and the down-regulated proteins were hydroxy-phenyllactic acid, creatinine, stearoylcarnitine, and inosine”. None of these are proteins! This is very basic and unacceptable for a research publication.
Response 2: The error description has been revised. In line 353&356
Comment 3-- Although the “intervention on alcoholic liver injury” is stated in the Title, no protective effects of CMAP were investigated in this study, but only the metabolomics on alcohol-treated L-02 hepatocytes.
Response 3: The correct descriptive statement has been selected. In line 1-3.
Comment 4-- The text itself is messy in general. For example, it is stated that two components (A1 and A2) were obtained and that A1 was “removed”. However, we see later in the text it was not “removed” but used for further studies. Besides, there are too many grammatical, spelling and other language-related errors (even in the title), which have to be corrected before submitting it to another journal.
Response 4: The error description has been corrected. In line 19.
Comment 5-- The Introduction contains only four references. The role of ACE inhibition in alcoholic liver disease (ALD) is not clear in this text. Besides, the abbreviation ALD is stated without explanation. Data on Nrf2 and p62 expression are not referred. In other parts of the manuscript, the references are missing too. It should be Lieber-DeCarli instead of ‘Liber-DeCari’ diet.
Response 5: Relevant information has been added. In line 308-347.
Comment 6-- In Methods section, several descriptions are missing – how the first extraction was performed, how ACE inhibition was assessed, which methods were used for antioxidant assays etc. It is stated that “approximately 200 μg/mL active peptides” were used in the cell treatment – why approximately? What is WB group?
Response 6: The error description has been corrected. In line 177. Group E and group D have the same materials, but different treatment methods. WB group is used to distinguish the two groups.
Comment 7-- In Results, there are parts of Discussion as well, although it should be separated in the manuscript (e.g. line 175). In lines 195-196 IC50 values are mentioned but it is not explained for which assay. There is IC50 value also for ‘Theragra chalcogramma Pallas’ but there is no reference for that.
Response 7: Correct modifications have been made. In line 126.
Comment 8-- In Figures 2 and 4, there are no explanations in the figure legends what is presented at figure parts A, B, C and D. Besides, what are the lowercase letters a, b, c in the figures? At Figure 4D, clearance rate (%) is marked at y-axis and the values are less than 1. The legend of Figure 5 is also inaccurate. The purification is a process and this is a figure that represents something else.
Response 8:Correct modifications have been made. In line 283&285.
Reviewer 3 Report
The manuscript presents interesting data on the isolation of active fraction of Camellia vietnamenis extracts and the effects of this fraction on the metabolome of L-02 cells subjected to the action of 200 mM ethanol. While the results are interesting, their non-professional presentation in the manuscript precludes, in my opinion, acceptance of the manuscript for publication in the present form.
Introduction: the first paragraph, with details of the isolation procedure, is not relevant.
It is not explained why inhibition of ACE is important in the context of prevention of ethanol effects.
If one of the aims of the manuscript is to “increase the comprehensive price of the Camellia oleifera industry”, the authors cannot declare that they have no conflict of interest.
Materials and Methods
Assay of ACE inhibition is not presented. The antioxidant methods are presented too shortly and partly wrongly (e.g. Ref. 9 does not describe method of assay of the “removal capacity of O2-“; instead of Ref. 8, the paper of Halliwell et al. should be cited.
Producers of the reagents used are not given.
Results
It is impossible to understand Figures 2 and 4 with the poor legend given – what is the “Clearance rate” (the same of three plots) and “Absorbance”?
I do not see a loading plot in Fig. 6. What is “high stachylne”?
Fig. 7: The numbers on the plot are not explained.
The manuscript would require a thorough linguistic check and amendment.
References:
Please correct references 1 and 3 and provide range of pages whenever available.
Minor remarks:
Lines 23 and 260/261: the compounds listed are not proteins
Line 51: improper word separations
Line 53: peroxiredoxin
Lines 55-56: wrong names
“.[1]” etc., period after citation, please
Author Response
Comment 1-- the first paragraph, with details of the isolation procedure, is not relevant.
It is not explained why inhibition of ACE is important in the context of prevention of ethanol effects.
If one of the aims of the manuscript is to “increase the comprehensive price of the Camellia oleifera industry”, the authors cannot declare that they have no conflict of interest.
Response 1: Correct modifications have been made. In line 30-36.
Comment 2-- Assay of ACE inhibition is not presented. The antioxidant methods are presented too shortly and partly wrongly (e.g. Ref. 9 does not describe method of assay of the “removal capacity of O2-“; instead of Ref. 8, the paper of Halliwell et al. should be cited.
Producers of the reagents used are not given.
Response 2: Related information has been added. In line 118-149.
Comment 3-- It is impossible to understand Figures 2 and 4 with the poor legend given – what is the “Clearance rate” (the same of three plots) and “Absorbance”?
I do not see a loading plot in Fig. 6. What is “high stachylne”?
Response 3: Correct modifications have been made. In line 307.
Comment 4-- Fig. 7: The numbers on the plot are not explained.
The manuscript would require a thorough linguistic check and amendment.
Response 4: Relevant missing data have been added. In line 308-347.
Comment 5—Please correct references 1 and 3 and provide range of pages whenever available.
Response 5: Related information has been added. In line 445.
Comment 6—Lines 23 and 260/261: the compounds listed are not proteins
Line 51: improper word separations
Line 53: peroxiredoxin
Lines 55-56: wrong names
Response 6: Mistakes in wording in the article have been corrected.
Round 2
Reviewer 1 Report
The authors have answered most of my questions and made corrections to the manuscript.
Although, the authors claim that "Relevant missing data have been added. In line 308-347 " in Response 11 to my comment, it is still unclear what authors try to illustrate with the two Venn diagrams in Figure 9 (former Figure 7). The Venn diagram uses overlapping circles to illustrate the logical relationships between sets of items. The first one (I suppose it is for up-regulated metabolites - there is still no legend in the figure) shows one common metabolite to all of the four sets. The authors claim there are two common metabolites. One of them - ribothymidine is not presented in figure 8. The second diagram (maybe representing down-regulated metabolites) shows four common metabolites. Two of them hydroxyphenyllactic acid, and inosine are not presented in figure 8. If there is not enough space for the representation of all of the metabolites, authors can add the information as Supplementary material, as I have mentioned in comment 10.
Author Response
Comment 1--- The authors have answered most of my questions and made corrections to the manuscript.
Although, the authors claim that "Relevant missing data have been added. In line 308-347 " in Response 11 to my comment, it is still unclear what authors try to illustrate with the two Venn diagrams in Figure 9 (former Figure 7). The Venn diagram uses overlapping circles to illustrate the logical relationships between sets of items. The first one (I suppose it is for up-regulated metabolites - there is still no legend in the figure) shows one common metabolite to all of the four sets. The authors claim there are two common metabolites. One of them - ribothymidine is not presented in figure 8. The second diagram (maybe representing down-regulated metabolites) shows four common metabolites. Two of them hydroxyphenyllactic acid, and inosine are not presented in figure 8. If there is not enough space for the representation of all of the metabolites, authors can add the information as Supplementary material, as I have mentioned in comment 10.
Response 1: Supplementary material has been added, and the presentation of detection methods such as DPPH has been revised.
Reviewer 2 Report
There is a slight improvement in the manuscript quality. However, several comments remained unanswered. On the other hand, some parts that are added in revised version of the manuscript are also confusing. For example, the text in lines 88-91 is in the form of instructions ("take", "immerse", "add") instead of describing the procedure. Besides, there are no adequate titles of the Method sections (ACE inhibition assay is in the "Ultrafiltration separation of crude Camellia vietnamensis active peptides" section). Some new results are added (PCA and clustering), although there is no information about these analyses in Methods.
Author Response
Comment 1-- There is a slight improvement in the manuscript quality. However, several comments remained unanswered. On the other hand, some parts that are added in revised version of the manuscript are also confusing. For example, the text in lines 88-91 is in the form of instructions ("take", "immerse", "add") instead of describing the procedure.
Response 1: The expression of the detection method has been modified. In line 91-102.
Comment 2--Besides, there are no adequate titles of the Method sections (ACE inhibition assay is in the "Ultrafiltration separation of crude Camellia vietnamensis active peptides" section).
Response 2: The title has been modified. In line 84-85.
Comment 3—Some new results are added (PCA and clustering), although there is no information about these analyses in Methods.
Response 3: Qualitative and quantitative analysis of metabolites by LC-MS followed by data analysis by software. This information has been presented in the paper. In line 196&212-216.

Reviewer 3 Report
The manuscript has been considerably improved but there are details, which still require amendment.
The title still requires amendment. Perhaps: “Preparation of active peptides from Camellia vietnamensis and Their Metabolic Effects in Alcohol-induced Liver Injury Cells”?
The sources of reagents have been given but the standard of such reporting is the includes the town and state of the supplier (e. g., “were purchased from Nanjing Jianghua Glass Instrument Co., Ltd. (Nanjing, China)”.
Line 93: “at 4 kR/min”; what is “kR”? My first association is kiloRoentgen; apparently the Authors meant “krpm”.
Lines 119-122: “take 0.5 mL of 1 mg/mL sample solution, add 2.5 mL of 0.1 mmol/L DPPH-absolute ethanol solution, leave it in the dark for 30 min, and detect the absorbance value at 517 nm, denoted”. The method should be presented as a report, not as an instruction. E.g.: “0.5 mL of 1 mg/mL sample was added to 2.5 mL of 0.1 mmol/L DPPH solution in absolute ethanol and decrease in absorbance at 517 nm was measured”.
Please indicate the time of incubation of the extract with DPPH solution (after which time after addition of the extract the absorbance was measured)?
Lines 127-133: “take 1 mL of 0.75 mmol/L phenanthroline solution prepared in absolute ethanol, add 2 mL of 0.15 mol/L phosphate buffer (pH 7.4) and 1 mL of 1 mg/mL sample solution, and immediately vortex to mix uniformly. Then, add 1 mL of 0.75 mmol/L ferrous sulfate solution, and, finally, add 1 mL of 0.1 mL/L H2O2 solution in Measure the absorbance value Ds and measure the ultrapure water instead of the absorbance value Dc of the sample and the H2O2 solution, and the absorbance value Db of the ultrapure water and the H2O2 solution instead of the sample” please change into, e.g.: “1 mL…. was added with 2 mL of….and 1 mL of…, and vortexed immediately… Then 1 mL of … ed and, finally, 1 mL of… was added and the sample was incubated in a water bath at 37 °C for 60 min. Then, absorbance at a wavelength of 536 nm was measured….”
Lines 134/135: “The H2O2 solution was determined”. Is this sentence necessary?
Lines 138-141: “Then, homogenize it and heat it in a water bath at 25 °C for 10 min. Following this, add 0.1 mL of 3 mmol/L pyrogallol solution, mix thoroughly immediately, measure the absorbance A at 325 nm at the same time, record the data every 30 s, and end at 5 min. “ This fragment should also be re-phrased e.g.:
“Then, the sample was homogenized and heated…. immediately mixed thoroughly and the absorbance (A) at 325 nm was measured. The data were recorded every 30 s for 5 min”
In particular, description of methods should be in the Past Tense, not as an instruction what to do.
Author Response
Comment 1-- The manuscript has been considerably improved but there are details, which still require amendment.
The title still requires amendment. Perhaps: “Preparation of active peptides from Camellia vietnamensis and Their Metabolic Effects in Alcohol-induced Liver Injury Cells”?
Response 1: The title has been modified. In line 2.
Comment 2-- The sources of reagents have been given but the standard of such reporting is the includes the town and state of the supplier (e. g., “were purchased from Nanjing Jianghua Glass Instrument Co., Ltd. (Nanjing, China)”.
Response 2: Related information has been added. In line 74-82.
Comment 3-- Line 93: “at 4 kR/min”; what is “kR”? My first association is kiloRoentgen; apparently the Authors meant “krpm”
Response 3: Correct modifications have been made. In line 95.
Comment 4-- Lines 119-122: “take 0.5 mL of 1 mg/mL sample solution, add 2.5 mL of 0.1 mmol/L DPPH-absolute ethanol solution, leave it in the dark for 30 min, and detect the absorbance value at 517 nm, denoted”. The method should be presented as a report, not as an instruction. E.g.: “0.5 mL of 1 mg/mL sample was added to 2.5 mL of 0.1 mmol/L DPPH solution in absolute ethanol and decrease in absorbance at 517 nm was measured”.
Please indicate the time of incubation of the extract with DPPH solution (after which time after addition of the extract the absorbance was measured)?
Lines 127-133: “take 1 mL of 0.75 mmol/L phenanthroline solution prepared in absolute ethanol, add 2 mL of 0.15 mol/L phosphate buffer (pH 7.4) and 1 mL of 1 mg/mL sample solution, and immediately vortex to mix uniformly. Then, add 1 mL of 0.75 mmol/L ferrous sulfate solution, and, finally, add 1 mL of 0.1 mL/L H2O2 solution in Measure the absorbance value Ds and measure the ultrapure water instead of the absorbance value Dc of the sample and the H2O2 solution, and the absorbance value Db of the ultrapure water and the H2O2 solution instead of the sample” please change into, e.g.: “1 mL…. was added with 2 mL of….and 1 mL of…, and vortexed immediately… Then 1 mL of … ed and, finally, 1 mL of… was added and the sample was incubated in a water bath at 37 °C for 60 min. Then, absorbance at a wavelength of 536 nm was measured….”
Lines 134/135: “The H2O2 solution was determined”. Is this sentence necessary?
Lines 138-141: “Then, homogenize it and heat it in a water bath at 25 °C for 10 min. Following this, add 0.1 mL of 3 mmol/L pyrogallol solution, mix thoroughly immediately, measure the absorbance A at 325 nm at the same time, record the data every 30 s, and end at 5 min. “ This fragment should also be re-phrased e.g.:
“Then, the sample was homogenized and heated…. immediately mixed thoroughly and the absorbance (A) at 325 nm was measured. The data were recorded every 30 s for 5 min”
In particular, description of methods should be in the Past Tense, not as an instruction what to do.
Response 4: The expression of the detection method has been modified. In line 120-141.